# Corporate Sustainable Development Strategy: Effect of Green Shared Vision on Organization Members’ Behavior

**DOI:** 10.3390/ijerph17072446

**Published:** 2020-04-03

**Authors:** Tai-Wei Chang

**Affiliations:** Graduate School of Resources Management and Decision Science, National Defense University, Taipei 112, Taiwan; taiwei661105@gmail.com

**Keywords:** green shared vision, green organizational identity, green product psychological ownership, organizational citizenship behavior for the environment

## Abstract

In this study, expectancy–valence theory was used from the perspective of corporate green management to investigate green shared vision (GSV). Moreover, organizational identity theory and psychological ownership theory were combined to propose an integrated conceptual framework. To fill research gaps, an investigation was also conducted with frontline R & D and sales employees to further examine the effect of GSV on organization members’ psychology and behavior. The research results indicate that when under the following circumstances, corporate members can enhance organizational citizenship behavior for the environment (OCBE) to improve green management performance, gain the favor of green consumers and achieve sustainable consumption goals. First, enterprises should focus on environmental issues and develop their GSV. Second, enterprises should implement the GSV in different aspects, such as product design, processing, marketing and management. Finally, corporate GSV should be in line with the expectations and values of organization members on environmental issues to motivate their OCBE. Therefore, enterprises must implement their GSV to strengthen the green organizational identity and green product psychological ownership of their members.

## 1. Introduction

With the increase in environmental issues, such as global climate change and the endangerment of animal species, high expectations are placed on enterprises to engage in behaviors for ecology maintenance. The contribution of corporate management to sustainable development has attracted widespread attention from scholars, managers and decision-makers [1,2,3,4,5,6]. Global warming and extreme climate influence the climate and ecosystem in many regions. These phenomena have triggered a wave of global environmental protection with increasingly stringent enactment of environmental protection laws and regulations, which has increased consumer awareness regarding environmental protection and has placed considerable pressure on business management. For instance, in terms of business orientation, Chinese paper enterprises have begun to invest in green management under the influence of internal and external pressure to improve their environmental behavior [7]. Furthermore, in 2015, Volkswagen was involved in a scandal related to the emissions of certain diesel-powered vehicles. An environmental pollution incident occurred in 2016 at Tinh Steel Mills, Vietnam, which was established by Taiwan’s Formosa Plastics Group. The aforementioned facts indicate that currently, the green shared vision (GSV) developed by enterprises does not involve the achievement of sustainable business goals when pursuing commercial interests. The aforementioned text also indicates that sustainable environmental requirements are increasingly stringent. The adoption of sustainable environmental management by enterprises would enable them to overcome environmental challenges, gain the favor of green consumers, achieve sustainable consumption goals and strengthen their corporate image and competitiveness.

In the past, a conflict existed between enterprise operating performance and environmental management, and environmental management was considered to cause an erosion of enterprise profits and competitiveness. However, environmental management, which affects all strategies and measures of an enterprise, has become a vital factor in the development of companies [8,9]. Environmental management transformation is a multidimensional and transdisciplinary process [10]. Hertin et al. [11] indicated that environmental management systems are the driving factor for improving environmental performance. Small and medium enterprises must also move toward green management through government incentives and informal peer pressure [12]. The strategic establishment of environmental public policies has affected the development of the green economy and has become a necessary condition for the development of sustainable environmental technology and the competitiveness of the green sector [13]. In other words, environmental management has become critical to business strategy [14]. With the popularization of global environmental protection and sustainable development, consumers have become increasingly conscious of the effect of the environment on living quality. Therefore, an increasing number of consumers are using green products to reduce environmental pollution and destruction. When selecting products, consumers consider environmental protection as an important standard of impulse and no longer consider price as the only purchase consideration [15,16,17,18]. Voluntary environmental behaviors in organizations drive high-performance environmental management [19] for the achievement of sustainable development goals (SDGs). However, if a company engages in pollution, environmental damage or greenwash behavior, its goodwill would be seriously affected, and it would experience substantial losses. Hence, green management performance is enhanced when business managers adopt an innovative and positive attitude. Creating a strategic vision is a crucial task for enterprise managers [20] because enterprise vision, which is the most powerful organizational tool [21,22], is the core of an enterprise’s overall business strategy and a critical factor for the development of leaders’ influence. From both personal and organizational perspectives, visions such as shared visions are the major driving forces for creating or enhancing the transition from current states to desired final states [23]. Shared vision is likely to mobilize participation in resource integration [24]. Consequently, the development of shared vision is an essential factor for an enterprise to obtain a competitive advantage. Currently, sustainable development has become a universal value worldwide and has evolved into an emerging industrial trend based on the achievement of SDGs, which have shaped the sustainable consumption model.

With regard to the implementation of a shared vision, the most important stakeholders are the employees of the organization. Felin et al. [25] considered employee involvement to be influential in the promotion of an enterprise’s environmental protection behavior. Moreover, they posited that interpersonal interaction among employees contributes to the promotion of sustainable management from a macro perspective. Studies have indicated that the effectiveness of green strategies depends on the degree to which employees take green actions automatically [26,27,28,29,30,31,32]. Considering that the environmental behavior of employees is conducive to the environmental protection achievements of an enterprise, many researchers have attempted to identify the factors influencing organizational citizenship behavior for the environment (OCBE). Moreover, the authors of Priyankara et al. [33] indicated that in the promotion of formal environmental management, employees’ OCBE can positively influence organizations if the organizations are faced with incomplete institutional norms or technical deficiencies. Although the initiative or voluntary green behavior of organization members is a detailed topic, with time, it will gradually become a crucial factor for an organization’s environmental performance [34,35,36,37,38,39]. Daily et al. [32] noted that OCBE accelerates the development of corporate green strategy. Therefore, OCBE is a crucial factor for achieving sustainability in green organizations [33]. Voegtlin et al. [40] stated that according to the role model effect, responsible leaders enhance the organizational citizenship behavior of employees. Being a responsible leader involves leading subordinates toward enterprise vision and goals. When an enterprise emphasizes, invests in and develops environmental protection strategies, economic, social and ecological benefits are integrated. Moreover, the GSV and goals of an enterprise are established. Consequently, an example is set for organization members to follow. Therefore, GSV contributes to the promotion of OCBE among employees.

The aforementioned discussion indicates that environmental management has become a crucial factor in business management [8,9,14]. Research indicates that the citizenship behavior of green organizations enhances the effectiveness of organizational green management, which allows organizations to gain the favor of green consumers [33,34,35,36,37,38,39]. Accordingly, improving the green management performance of enterprises is key to establishing a suitable sustainable consumption model. A shared vision enables the future development of green behavior in members [41,42] and makes members believe that their contributions are meaningful [43]. Therefore, we examined the importance of corporate GSV for green management. Related research on GSV has examined organization members’ green creativity [44], green radical innovation and green incremental innovation [45], reactive green innovation [46], pro-environmental behavior [47] and green product development performance [48,49]. Theories have rarely been used to study the aforementioned factors. This study draws on the perspectives of expectancy–valence theory, organizational identity theory and psychological ownership theory to develop a comprehensive model for investigating the effect of GSV on the behavior and psychology of the frontline sales and R & D personnel of enterprises. Expectancy–valence theory indicates that people are affected by expected values and benefits, which influences their behavior and performance. The effects of GSV on green organizational identity (GOI), green product psychological ownership (GPPO); OCBE were explored from the perspectives of the aforementioned three theories. To fill the gap between current research theories and environmental management issues, expand the literature on enterprise sustainable development, encourage enterprises to win the favor of green consumers and achieve the SDGs of enterprises, we conducted empirical tests to verify the relationships among the aforementioned variables.

To explore the relationships between the aforementioned factors, the following steps were followed. First, we summarized the relevant literature on GSV, GOI, GPPO and OCBE. Next, we conducted empirical tests to verify the relationships between these important variables. Specifically, eight proposed hypotheses were examined according to the research framework. Finally, we obtained conclusions after examining the research findings and implications.

## 2. Literature Review and Hypothesis Development

### 2.1. Influence of GSV on Organization Members According to Expectancy–Valence Theory

Under increasingly severe environmental challenges, ideology affects the promotion of environmental policies [50]. In response to the rise in environmental awareness, the concept of environmental protection has been gradually integrated into discussions of green management. Because organizational identification helps organization members to understand how their work and tasks are related to their organization, enterprises should not ignore green management issues when environmental issues play a vital role in organizational identity [49]. GOI refers to the organizational identity of members when an organization is under an interpreted framework of environmental management and protection, whose structure is jointly constructed by organization members and which gives importance to the behaviors of its members [49,51]. Thus, a major difference between GOI and organizational identification is that GOI focuses on issues related to environmental management and protection, whereas organizational identification does not focus on these issues. Consequently, GOI has the characteristics and attributes of organizational identity. Geraie and Rad [52] stated that in current society, which attaches considerable importance to environmental issues, GOI is conducive to maintaining competitive advantages and increasing the quality and quantity of organizational innovation and competitiveness. Studies have indicated that factors such as individual values and behaviors [53], articulation of a vision [54], perceived organizational support [55], employee corporate social responsibility involvement [56], psychological empowerment [57] and organizational justice [58] shape or reinforce organizational identity. According to relevant research, shared vision represents the organizational vision of organization members and the collective concept of the organization’s mission and core values [59,60]. When organizational planners have strong visions, the achievement of sustainability requirements is promoted [61]. Consequently, shaping and clarifying the shared vision of an organization and creating personal value for organization members are key to generating organizational identity. GSV is positively related to GOI; thus, the following hypothesis is proposed:

**Hypothesis 1** **(H1).**
*GSV is positively associated with GOI.*


When environmental management is adopted by the internal organizations of enterprises, organization members are encouraged to make increased commitments to environmental activities [62]. Achieving suitable environmental performance depends on the establishment of an appropriate shared vision between managers and employees, especially those intensively participating in environmental strategies [63]. GSV indicates that managers set strategic directions and goals for environmentally friendly and sustainable development [45,49]. According to Vroom [64], expectancy–valence theory indicates that when personal effort input increases, positive results are achieved, which leads to people making additional efforts to achieve more valuable results. According to expectancy–valence theory, the selection, performance and sustainability of achievement behavior is influenced by expectations and value. If expected performance and value directly affect the choice of achievement and are affected by specific beliefs such as ability beliefs, then task difficulty, personal goals, performance, workload and persistence are influenced by expected performance and value [65,66,67,68,69]. A key factor affecting the success of green organizations is the behavior of their members toward the environment [32,37,39,49,70,71,72]. Schild [73] stated that personal environmental citizenship behavior is incentivized by environmental values. Developing a clear GSV is helpful for enhancing the OCBE of organization members [49]. OCBE refers to the spontaneous behavior of organization members, which is not included in the reward and assessment system, for improving the organization’s green management [31,32,49,74]. Piccolo and Colquitt [75] argued that organization members with high GOI are willing to give up their personal interests to achieve organizational goals and make positive contributions to organizations. When an individual’s unpaid contribution to the organization is combined with their environmental spirit, the organization can implement environmental management more effectively despite the fact that these behaviors are an informal reward [31]. Hence, according to expectancy–valence theory, when the GSV formulated by the enterprise provides the value expected by group members, OCBE is enhanced [49]. Consequently, the following hypothesis is proposed.

**Hypothesis 2** **(H2).**
*GSV is positively associated with OCBE.*


When psychological ownership arises, individuals regard their property as an extension of themselves [76,77,78,79] and their inner feelings [80]. When property is psychologically accepted by an individual, the property becomes “theirs” because they exist within it and merge with it [81]. Factors influencing the formation of psychological ownership are not confined to psychological ownership and job characteristics; they also include organizational culture, the attitude of senior managers, corporate goals, vision and reputation [82]. According to Jussila et al. [83], the psychological ownership of individuals originates from the exercise of control, acquisition of intimate knowledge and self-investment and is affected by efficacy and effectance, self-identity, having a place and stimulation. Thus, multiple factors influence psychological ownership; these factors are dependent on specific subjects. The concept of psychological ownership indicates that goals are one feature of psychological ownership identity, whose core refers to sense of ownership and the psychological relevance to the subject matter. The concept encompasses the total or partial ownership feelings of an individual (i.e., “It is mine.”). The sense of psychological ownership may vary with different objects [84]. Thus, organization members can have psychological ownership of a company’s green products and a strong sense of ownership of the subject matter. Moreover, they may have positive attitudes and may be willing to engage in extra-role behaviors and take risks, invest efforts and make sacrifices for the organization. This phenomenon can be referred to as GPPO.

Studies on expectancy–valence theory have indicated that psychological ownership, individual values and behaviors [53]; vision articulation [54] are beneficial to organizational identity. When psychological ownership is generated in an organization member, the member regards a subject matter as an extension of the self [76,77,78,79]. Psychological ownership is affected by self-identity motivation [83] as well as corporate goals and vision [82]. Consequently, organization members have psychological ownership of the company’s green products. The aforementioned characteristics cause GPPO to have several effects; thus, the following hypotheses are proposed:

**Hypothesis 3** **(H3).**
*GOI is positively associated with GPPO.*


**Hypothesis 4** **(H4).**
*GSV is positively associated with GPPO.*


**Hypothesis 5** **(H5).**
*GOI mediates the relationship between exposure to GSV and GPPO.*


### 2.2. Influence of GPPO on Organization Members According to Psychological Ownership Theory

Ownership is a pluralistic aspect that is both objective and a representation of mental experience. Ownership refers to the feeling of an individual that the whole or part of a subject matter belongs to them (i.e., It is “mine” or It is “ours”). Employees reflect their self-form (functions of form and psychology) in their ownership attitude or behavior [85]. Psychological ownership influences an individual’s attitude, motivation and behavior and is characterized by attitude, self-identity and a sense of responsibility [86]. It can also give rise to out-of-role behavior [87], namely organizational citizenship behavior. If individuals have psychological ownership of something, they are willing to take risks or make personal sacrifices for it [88]. Accordingly, the psychological ownership of organization members brings the organization and its employees closer and equips employees with positive attitudes and ideas, which improve their work performance [89]. Studies on psychological ownership related to organizations have revealed that if employees have psychological ownership of their organizations, their job satisfaction and job performance can be enhanced. Job satisfaction and performance can be predicted by the degree of organizational psychological ownership of members [84,88,90].

In conclusion, when organization members have psychological ownership of a company’s green products, they pay more attention to green products, take measures spontaneously to improve green management (i.e., OCBE). Therefore, the following hypothesis is proposed:

**Hypothesis 6** **(H6).**
*GPPO is positively associated with OCBE.*


Pierce et al. [85] indicated that the psychological ownership of employees is affected by access to the organization, access to the information of the organization and conditions for the lawful exercise of influence. These three factors also affect employees’ feelings, attitudes and behavioral responses. The psychological ownership of employees also influences cooperation, norms of working groups and peer pressure. Ownership enables the owner to think of a subject matter as a social entity because it establishes a psychological link between the subject matter and its owner. If a psychological connection exists between work and emotion, an organization and its employees will be intimately linked and employees will have positive attitudes and ideas toward the organization, which would enhance their performance [89]. The authors of Jussila et al. [83] suggested that motivation affects the status of individual psychological ownership, the generation of proactive attitudes (attitudinal consequences) and behaviors (behavioral consequences). According to Baxter et al. [91], psychological ownership is influenced by motivations and goals, encouraging or arousing psychological ownership and seeking actions, which in turn achieves attached motivation and goals. Thus, psychological ownership can be used as a medium between motivation and goals, as indicated by relevant studies [90,92,93,94,95,96].

According to social identity theory organizational identity is a form of social identity [97] that refers to the identity feelings of an individual in an organization and their internalization of organizational characteristics. These characteristics may still exist in the individual even after they leave the organization [98]. Organizational identity comprises similarity, membership and loyalty and promotes the sharing of solidarity, support and trust and goals or experiences with other members [99]. The more an individual perceives themselves from the organization’s perspective, the more they identify with the organization and the more their attitudes and behaviors are affected by their relationships with the organization members [100]. According to Porteous [101]; Pierce et al. [84]; Van Dyne and Pierce [86]; Karahanna and Zhang [102]; Jussila et al. [83], when psychological ownership of an object occurs in an individual, the individual feels that the object is theirs (“it is mine”) and they are willing to take risks to maintain the ownership. This psychological ownership is driven by the self-identification and attraction toward the object, which cause individuals to have positive attitudes and behaviors. When an individual has a sense of ownership of a subject matter and is willing to take risks to maintain the ownership, the psychological ownership is driven by self-identification and attraction toward the subject matter, which is manifested in the individual’s positive attitudes and behaviors.

The connotation of organizational identity is consistent with the antecedent characteristics of psychological ownership because organizational identity positively influences organizational citizenship behaviors [103,104,105,106,107]. GOI, GPPO and OCBE focus on issues related to environmental management and protection. Furthermore, Chang et al. [49] indicated that GOI positively affects OCBE; therefore, these factors have unique attributes. According to organizational identity theory and psychological ownership theory, when group members internalize the features of an organization into themselves, sense of ownership, namely GPPO, is strengthened to a certain extent and influences OCBE. Consequently, the following hypothesis is proposed:

**Hypothesis 7** **(H7).**
*GPPO mediates the relationship between exposure to GOI and OCBE.*


At the organizational level, a company’s goals and vision influence the psychological ownership of its members [82] and cause them to engage in active behaviors. The job satisfaction organizational citizenship behavior and performance of employees is also influenced by a company’s goals and visions [95,108]. If a shared vision is developed appropriately, employees would be provided goals and value, which is conducive for the improvement of their behavior, job satisfaction and performance. The entire process of motivation is in line with expectancy–valence theory [65,66,67,68,69]. According to expectancy–valence theory and psychological ownership theory, when organizations implement a GSV, which provide strategic direction for the growth of environmental protection initiatives, provide members with goals and value and strengthen the psychological sense of possession for green products (GPPO), which enhances their members’ OCBE. Hence, the following hypothesis is proposed.

**Hypothesis 8** **(H8).**
*GPPO mediates the relationship between exposure to GSV and OCBE.*


According to the aforementioned hypotheses and related studies, we designed the research framework, which considers the effects of GOI, GSV, GPPO and OCBE (Figure 1).

## 3. Methodology and Measurement

### 3.1. Data Collection and Sample

In this study, data were collected through a questionnaire survey to verify the aforementioned eight hypotheses. The research participants comprised employees of major enterprises in Taiwan. According to the statistics of the Bureau of International Trade, Ministry of Economic Affairs, R.O.C., electrical equipment and parts as well as machinery equipment and parts accounted for approximately 55% of the total exports in Taiwan in 2019. Electrical equipment and parts include integrated circuits; storage devices for discs, tapes and nonvolatile solids; diodes; transistors; congeneric semiconductor devices; and printed circuits. Machinery equipment and parts include automatic data processors and their affiliated parts as well as parts or accessories used mainly in machines with headings of 8470 to 8472. Companies specializing in eight industries, namely those for electronics, information services, component manufacturing, computer and peripheral products, electronic products and components, communication equipment manufacturing, machinery and equipment manufacturing and software, were selected randomly from the Taiwan Enterprise Directory. The respondents were the company’s formal frontline R & D and sales employees. To improve the valid response rate, reliability and validity of the questionnaire, this study was conducted under the guidance of relevant personnel and the respondents were provided with gifts. Moreover, a questionnaire survey was conducted by post to verify the proposed hypothesis. A total of 786 formal questionnaires were issued; 474 valid questionnaires were collected; thus, the effective response rate was 60.305%. The proportion of participants belonging to each industry was as follows: electronics, 12.93%; information services, 11.85%; component manufacturing, 12.72%; computer and peripheral products, 25%; electronic products and components, 14.87%; communication equipment manufacturing, 9.7%; machinery and equipment manufacturing, 6.9%; and software industries, 6.03%. The size of staff in the surveyed companies was as follows: fewer than 100 employees, 165 companies (34.81%); more than 100 but less than 500 employees, 193 companies (40.72%); more than 500 but less than 1000 employees, 98 companies (approximately 20.68%); and over 1000 staff members, 3 companies (approximately 3.8%) (Table 1). Thus, mainly small- or medium-sized companies were surveyed in this study.

### 3.2. Measurements

The questionnaire used in this study contained four measurement dimensions, namely GSV, GOI, GPPO and OCBE. The questionnaire was developed according to the advice of relevant scholars (Appendix A), and the questionnaire items were rated on a 7-point Likert scale ranging from 1 for strongly disagree to 7 for strongly agree. The respondents rated the items by ticking relevant checkboxes. The reliability coefficients for GSV, GOI, GPPO and OCBE were 0.897, 0.936, 0.847 and 0.964, respectively. These values are higher than the standard value of 0.7 proposed in Nunnally and Bernstein [109]; thus, the adopted questionnaire had high reliability.

## 4. Empirical Results

### 4.1. Measurement Model Results

Table 2 presents the means, standard deviations, correlations and square roots of average variance extracted (AVE) of the constructs. The *p*-value reaches a significant standard, and Table 2 shows the significant correlation between the variables.

Table 3 presents the factor analysis results for GSV, GOI, GPPO and OCBE. The results for each construct are presented in terms of the accumulation percentage of explained variance.

Table 4 presents the loading (λ) values for the four constructs. As presented in Table 4, all the factor loadings were higher than 0.7; thus, the questionnaire had a suitable individual item reliability [110]. Moreover, all the composite reliability (CR) and Cronbach α values were above 0.7, which indicated that the questionnaire had suitable internal consistency reliability [111]. On the basis of Fornell and Larcker [112], AVE was adopted in this study to measure discriminant validity. The considered constructs have suitable discriminant validity only when the square roots of AVE for each construct are higher than the correlation coefficients among the constructs. Moreover, if the AVE value is higher than 0.5, the construct has suitable convergent validity. The CR of GSV, GOI, GPPO and OCBE were 0.898, 0.936, 0.845 and 0.965, respectively, which are above 0.5; thus, the four constructs had satisfactory convergent validity. Table 3 indicates that the square roots of AVE of GSV, GOI, GPPO and OCBE were 0.83, 0.843, 0.803 and 0.855, respectively. All the AVEs were higher than the corresponding correlation coefficients; thus, the four constructs had appropriate discriminant validity. Moreover, the AVEs (0.689, 0.710, 0.645 and 0.732) listed in Table 4 are higher than 0.5, which indicates that the four constructs had satisfactory convergent validity.

The aforementioned results indicate that the questionnaire used in this study had acceptable reliability and validity.

### 4.2. Structural Model Results

In this study, structural equation modeling was performed using Amos 26 statistical analysis software to analyze and verify the proposed hypotheses. Figure 2 displays the verification results of the full model, which indicate that adapting full model is acceptable (chi square/df = 2.928, GFI = 0.886, RMSEA = 0.064, NFI = 0.956 and CFI = 0.956). The five direct paths in the full model exhibited significant positive effects (All *p*-values are reaching significant standards.); the results indicate that H1, H2, H3, H4 and H6 are valid.

According to the research results, an increase in GSV can enhance GOI, GPPO and OCBE; thus, GSV is an important driving force for the aforementioned three factors. In addition, organizations must enhance their members’ OCBE such that GSV, GOI and GPPO can be improved. Through adopting the mediating effect test presented in Baron and Kenny [113], three mediation paths were identified in the structural model. In model 1, GOI is a mediator between GSV and GPPO; in model 2, GPPO is a mediator between GOI and OCBE; and in model 3, GPPO is a mediator between GSV and OCBE. Table 5 indicates that the aforementioned three mediation paths in the structural models were suitable and acceptable.

As presented in Table 5, the model fit the indices and the path coefficient of each model generated statistically significant effects. In model 1, GSV has a positive correlation with GOI, GSV and GPPO, which proves the mediating role of GOI. Thus, an organization must enhance GPPO by promoting GSV and GOI. In model 2, GOI has a direct positive correlation with GPPO and OCBE and GPPO has a direct positive correlation with OCBE; thus, GPPO acts as a mediator in the relationship between GOI and OCBE. An organization can modify GOI and GPPO to strengthen the OCBE of members. In model 3, GSV has a positive correlation with GPPO and OCBE, and GPPO has a positive correlation with OCBE, which confirms the intermediary effects of GPPO in the relationship between GOI and OCBE. Therefore, an organization should improve GSV and GPPO to enhance the OCBE of members.

The aforementioned results indicate that GSV, GOI and GPPO are vital influence factors for the OCBE of organization members.

To ensure preciseness, percentile bootstrap and deviation correction percentile bootstrap were performed with 5000 samples and a 95% confidence interval in examining the three mediation models [114]. An examination of the indirect effects of the variables confirmed the three intermediary relationships. In the mediating model, we focused on not only the overall indirect effect of shared vision on OCBE (0.857, z = 6.436, *p* < 0.001) but also on the specific indirect effects. The specific indirect effects of the path model were as follows: (1) point estimation = 0.224 (z = 4.978, *p* < 0.001) for the mediating effect of GOI on the relationship between GSV and GPPO; (2) point estimation = 0.0336 (z = 5.015, *p* < 0.001) for the mediating effect of GPPO on the relationship between GOI and OCBE; and (3) point estimation = 0.0159 (z = 3.383, *p* < 0.001) for the mediating effect of GPPO on the relationship between GSV and OCBE. We continued to collect relevant data to verify the relationships between GSV, GOI, GPPO and OCBE, and accidentally discovered a fourth mediation model, whose result can be represented as follows: (4) point estimation = 0.0138 (z = 4.182, *p* < 0.001) for the mediating effects of GOI and GPPO on the relationship between GSV and OCBE. The fourth model had distal indirect effects. The sum of all the specific indirect effects of the path model was 0.857.

We must determine whether 0 is within the 95% confidence interval because when 0 is included in the interval, no mediating effect exists [115]. As presented in Table 6, 0 did not occur between the lower and upper limits of each indirect effect. Therefore, the mediating effect was significant. Consequently, hypotheses 5, 7 and 8 were confirmed. Specifically—according to the research results of the intermediary model constructed in this study—GOI and GPPO have mediating effects because their 95% CI values do not contain 0. An examination of the paired comparison of indirect effects indicated no obvious difference in the indirect effects of GPPO and GOI; however, a significant difference was noted in the mediating effects of GPPO and OCBE under different antecedents of GOI and GSV.

In summary, all the proposed hypotheses were confirmed (All *p*-values and or z-values are reaching significant standards which presented in Table 7). Moreover, analysis of the collected data indicated that GSV has distal indirect effects and can exert a positive influence on OCBE through GOI and GPP.

## 5. Conclusions and Implications

In the era of rising consumer environmental awareness, enterprises play an important role in moving toward a sustainable consumption model. Moreover, the implementation of sustainable development and the acquisition of consumer favor are crucial factors for achieving sustainable consumption. With the development of a green economy, the active pursuit of high environmental protection standards would bring profits to enterprises [116]. Research has indicated that when consumers who value environmental protection make a purchase decision, environmental factors are included in their assessments. Such consumers prefer green products over traditional products [117,118,119,120,121] and are willing to spend more money on green products [122]. Therefore, the sustainable development of an enterprise would improve its competitive advantage [9,123,124,125]. Environmental management is regarded as an essential business strategy for achieving sustainable development [14]. Considering the critical influence of sustainable enterprise development on sustainable consumption, this study was an attempt to explore the factors that affect OCBE by exploring the influence of GSV on the OCBE psychological process of first-line R & D and sales staff. The overall green performance of a company should be improved to meet the expectations of green consumers.

In this study, expectancy–valence organizational identity and psychological ownership theories were used to obtain an integrated conceptual framework. The framework indicates that the generation of GSV in organization members depends on whether the enterprise meets their expectations and values and whether it recognizes and produces a psychological behavior history relationship between GPPO and active behavior. The research results indicate that (1) GSV has positive effects on GOI, GPPO and OCBE; (2) GOI has a positive effect on GPPO; (3) GPPO has a positive effect on OCBE; (4) GOI plays a mediating role in the relationship between GSV and GPPO; (5) GPPO plays a mediating role in the relationship between GOI and OCBE; (6) GPPO plays a mediating role in the relationship between GSV and OCBE; and (7) GOI and GPPO play mediating roles in the relationship between GSV and OCBE. Enterprises should formulate and implement a clear, prospective and valuable GSV. When employees truly own the GSV, their aspirations are in line with those of the organization and their colleagues, which results in GOI interconnections and bundling. Consequently, the individual’s GPPO level is enhanced, which makes them more willing to actively participate in OCBE. The study results indicate that GSV formulation improves the overall green management performance of the enterprise, which allows the enterprise to attract green consumers and achieve a sustainable consumption model.

This study has four limitations. First, expectancy–valence theory does not explore the external or internal expected value motivation of GSV for organization members. This aspect can be examined in future studies. Second, the research samples focus on the company perspective and does not discuss demographic variables in detail. Future research can further verify include age, education, etc. Third, analysis of the collected research data indicates that GSV has distal indirect effects and can exert a positive effect on OCBE through GOI and GPP. Thus, further research must be conducted to examine the effect of remote mediation and verify the unexpected findings of this study. Finally, all the research participants were employees of Taiwanese companies. Differences may exist between different countries in terms of the national conditions, culture, industry, economic structure and national production income.

According to the research results, three management implications of OCBE are proposed. First, GSV is an indispensable factor for an enterprise that aims to improve the GOI, GPPO and OCBE of their members for winning the favor of green consumers and achieving sustainable consumption. Studies have indicated that vision is the core of enterprise management and a crucial factor for enhancing performance [21,22,24]. Therefore, to enhance OCBE, enterprises should first develop an understandable and valuable vision. For instance, managers must determine what their short-, medium- and long-term GSVs are as well as determine the purpose of formulating such GSVs. If such a strategy is adopted, employees are likely to understand and identify with the enterprise, which improves OCBE and overall green management performance, advances SDGs and enhances the competitiveness of the enterprise in the green economy market. Second, Temminck et al. [126] indicated that when employees have a high awareness of environmental issues, their OCBE improves. Hence, GSV is a crucial consideration for enterprises. To improve the GOI of enterprise members, an appropriate GSV should be implemented. Such a GSV should encompass the implementation of corporate social responsibility [127], the publishing of social and environmental reports [128], the enhancement of green training courses, energy saving, carbon emission reduction and the timely adjustment of green supply chains. If such a GSV is adopted and implemented by an enterprise, its employees at all levels would realize that GSV is not a mere slogan, which would improve their GOI. Finally, enterprises must cooperate with human resource management to establish relevant rewards and punishment as well as promotion and training regulations. Moreover, special praise should be given to employees who not only finish due tasks but also participate in additional work for improving enterprise performance. Such praise would enhance the GOI, GPPO and OCBE of the employees, which would improve the enterprise’s environmental performance. In addition, a direct emphasis on adopting green human resource management practices would elevate the environmental performance of enterprises [129].

This paper proposes five directions for further research. First, GOI and GPPO play mediating roles in the relationship between GSV and OCBE. Moreover, GSV has distal indirect effects and exerts a positive effect on OCBE through GOI and GPPO. This finding can be further verified by examining relevant literature and theories. Second, the research framework focuses on the formation of OCBE. Future studies can consider the internal environmental factors of an organization as an entry point and focus on GSV, GOI and GPPO. Moreover, researchers can incorporate other organizational internal factors, such as leadership style, unsuitable supervision and workplace gossip or external factors into their study. Third, a quantitative method was adopted in this study. In the future, both qualitative and quantitative methods can be used to further understand the influence factors of the OCBE of organization members. Fourth, the research sample can be widened to include listed companies or companies across different countries to explore how the results compare with those of this study. Finally, the influence of demographic variables, such as gender, marriage, education, length of work and position, can be explored.

## Figures and Tables

**Figure 1 ijerph-17-02446-f001:**
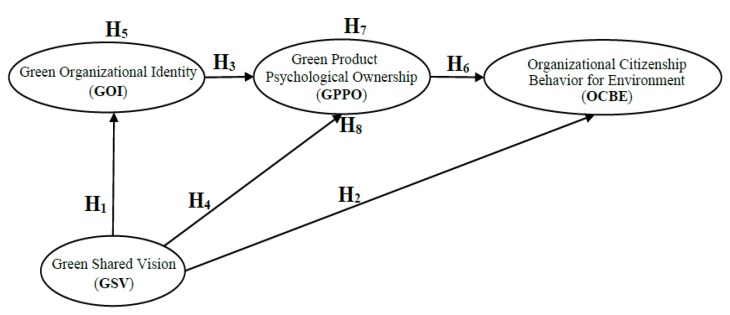
Research framework.

**Figure 2 ijerph-17-02446-f002:**
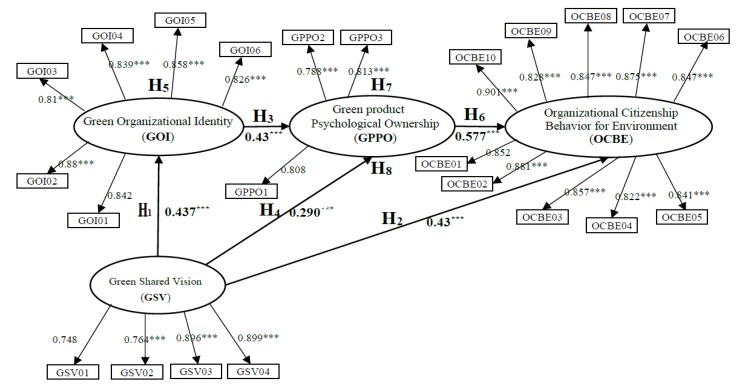
Results of the full model. chi square/df = 2.928, GFI = 0.886, RMSEA = 0.064, NFI = 0.956, CFI = 0.956. Note: *** *p* < 0.001.

**Table 1 ijerph-17-02446-t001:** Sample distribution by industry classification.

Industry	Number of Samples	Percent of Sample (%)	Size of Firm	Number of Samples	Percent of Sample (%)
electronics	60	12.93%	less than 100 people	165	34.81%
information services	55	11.85%			
components manufacturing	59	12.72%			
computer and peripheral products	116	25%	100–500	193	40.72%
electronic products and components	69	14.87%			
communication equipment manufacturing	45	9.7%			
machinery and equipment manufacturing	32	6.9%	500–1000	98	20.68%
software industries	28	6.03%	more than 1000 people	18	3.8%
Total	474	100%	Total	474	100%

**Table 2 ijerph-17-02446-t002:** Means, standard deviations and correlations of the constructs.

Constructs	Mean	Standard Deviation	A	B	C	D
A. GSV	4.794	1.031	(0.83)			
B. GOI	5.232	0.821	0.385 **	(0.843)		
C. GPPO	5.188	0.848	0.469 **	0.613 **	(0.803)	
D. OCBE	4.581	1.019	0.629 **	0.547 **	0.624 **	(0.855)

Notes: (1) Diagonal elements are the square roots of AVE; (2) **: *p* < 0.01. GSV, green shared vision; GOI, green organizational identity; GPPO, green product psychological ownership; OCBE organizational citizenship behavior for the environment.

**Table 3 ijerph-17-02446-t003:** Factor analysis results for the constructs considered in this study.

Constructs	Number of Items	Number of Factors	Accumulation Percentage of Explained Variance
GSV	4	1	76.46%
GOI	6	1	75.777%
GPPO	3	1	76.679%
OCBE	10	1	75.851%

Note: GSV, green shared vision; GOI, green organizational identity; GPPO, green product psychological ownership; OCBE organizational citizenship behavior for the environment.

**Table 4 ijerph-17-02446-t004:** Item loadings (λ) and Cronbach α coefficients and AVEs of the constructs.

Constructs	Item Number	Factor Loading	Cronbach’s α	CR	AVE	$\sqrt{\mathrm{AVE}}$
GSV	GSV01	0.748	0.897	0.898	0.689	0.83
	GSV02	0.764 ***				
	GSV03	0.896 ***				
	GSV04	0.899 ***				
GOI	GOI01	0.842	0.936	0.936	0.710	0.843
	GOI02	0.880 ***				
	GOI03	0.810 ***				
	GOI04	0.839 ***				
	GOI05	0.858 ***				
	GOI06	0.826 ***				
GPPO	GPPO1	0.808	0.847	0.845	0.645	0.803
	GPPO2	0.788 ***				
	GPPO3	0.813 ***				
OCBE	OCBE01	0.852	0.964	0.965	0.732	0.855
	OCBE02	0.881 ***				
	OCBE03	0.857 ***				
	OCBE04	0.822 ***				
	OCBE05	0.841 ***				
	OCBE06	0.847 ***				
	OCBE07	0.875 ***				
	OCBE08	0.847 ***				
	OCBE09	0.828 ***				
	OCBE10	0.901 ***				

Note: *** *p* < 0.001.

**Table 5 ijerph-17-02446-t005:** Results of the structural mediation models.

Result		Model 1	Model 2	Model 3
Model fit	Chi square/df	2.384	3.201	3.447
	GFI	0.953	0.902	0.907
	RMSEA	0.054	0.068	0.072
	NFI	0.967	0.944	0.946
	CFI	0.981	0.961	0.961
Path Coefficient		GSV → GOI 0.431 ***	GOI → GPPO 0.688 ***	GSV → GPPO 0.543 ***
		GOI → GPPO 0.557 ***	GPPO → OCBE 0.554 ***	GPPO → OCBE 0.448 ***
		GSV → GPPO 0.302 ***	GOI → OCBE 0.197 ***	GSV → OCBE 0.441 ***

Notes: (1) ***: *p* < 0.001. (2) Model 1, GSV → GOI -> GPPO; Model 2, GOI → GPPO → OCBE; Model 3, GSV → GPPO → OCBE.

**Table 6 ijerph-17-02446-t006:** Mediation results of GPPO and OCBE when using a confidence interval bootstrap.

Path	Point Estimation	Product of Coefficients		Bootstrapping			
				Bias-Corrected 95% CI		Percentile 95% CI	
		S.E.	Z	Lower	Upper	Lower	Upper
**Indirect Effects**							
(1) GSV → GOI → GPPO	0.224	0.045	4.978 ***	0.146	0.325	0.144	0.321
(2) GOI → GPPO → OCBE	0.336	0.067	5.015 ***	0.221	0.481	0.213	0.475
(3) GSV → GPPO → OCBE	0.159	0.047	3.383 ***	0.088	0.274	0.084	0.267
(4) GSV → GOI → GPPO → OCBE	0.138	0.033	4.182 ***	0.085	0.217	0.081	0.211
Total (1 + 2 + 3 + 4)	0.857	0.133	6.436 ***	0.620	1.146	0.616	1.143
**Contrasts**							
(1)−(2)	−0.112	0.079	−1.418	−0.28	0.033	−0.271	0.038
(2)−(3)	0.177	0.069	2.565 *	0.055	0.327	0.045	0.313
(3)−(1)	−0.065	0.067	−0.942	−0.199	0.065	−0.196	0.068

Notes: (1) Standardized estimation of 5000 bootstrap samples. (2) Differences in the two indirect effects. (3) GSV, green shared vision; GOI, green organizational identity; GPPO, green product psychological ownership; OCBE organizational citizenship behavior for the environment. (4) ***: Z > 3.29, *: Z > 1.96. (5) *N* = 474.

**Table 7 ijerph-17-02446-t007:** Results of the structural model.

Hypothesis	Path Coefficient	Z Value	Results
H1	0.437 ***		H1 is supported
H2	0.43 ***		H2 is supported
H3	0.43 ***		H3 is supported
H4	0.29 ***		H4 is supported
H6	0.577 ***		H6 is supported
mediates the relations			
H5	GSV → GOI → GPPO	4.978 ^###^	H5 is supported
H7	GOI → GPPO → OCBE	5.015 ^###^	H7 is supported
H8	GSV → GPPO → OCBE	3.383 ^###^	H8 is supported
Study found	GSV → GOI->GPPO → OCBE	4.182 ^###^	Distal indirect effect

Note: (1) ***: *p* < 0.001. (2) ^###^: Z > 3.29. (3) STE, standardized total effect; SIE, standardized indirect effect; SDE, standardized direct effect.

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
