# Peer review of "Corporate Sustainable Development Strategy: Effect of Green Shared Vision on Organization Members’ Behavior"

_ijerph, 2020, doi:10.3390/ijerph17072446_

Round 1

Reviewer 1 Report

This paper addresses a very interesting topic. However, this paper suffers a lot from its conceptual sections. I will elaborate on them largely according to the sequence of the paper. 

  1. Title: The title of this article is too long. I can not realize the content of this study through the title.
  2. The introduction lack of enlightenment on research background and problem statement. The author proposes three examples (Chinese paper enterprise, Volkswagen, and Formosa Plastics Group) of environmental management, but there are weak connections between research focus and the three examples.
  3. After reading the Intro, I have no idea about (1) research gaps and how they are identified, (2) overarching theory, (3) why the current study is necessary and important, and (4) where the current study makes a contribution to.
  4. The authors need to make a better positioning of their study, This study focuses on the theoretical perspectives of expectancy-valence theory, organizational identity theory, and psychological ownership theory. The author develops a comprehensive model to verify the direct and indirect effects of the GSV on green organizational identity (GOI), green product psychological ownership (GPPO), and OCBE. My suggestion is to rewrite this sentence and explores the relationship between each factor. Authors need to provide a better justification for conducting their study, as well as state clearly the purpose, objectives, and contributions of it. 
  5. My main concern is that the article presents nothing new. The final section provides only a summary of the findings than a fruitful discussion and insightful conclusions. In addition, there are no theoretical and managerial implications. The overall communication style is not good enough.
  6. I'd strongly suggest seeking editorial help as grammar and usage errors throughout the paper are limiting clarity. In the present form, the work is not good enough.

Author Response

The response to the Reviewer 1s  comments

Manuscript ID:

ijerph-740402

Title:

Corporate strategies for sustainable consumption: explore the significant motives and antecedents of organizational citizenship behavior for the environment in corporate environmental management

Corporate Sustainable Development Strategy: Effect of Green Shared Vision on Organization Members’ Behavior

【Reviewer 1’s Comment 1】:

【Authors’ Response】:

Comments and Suggestions for Authors

This paper addresses a very interesting topic. However, this paper suffers a lot from its conceptual sections. I will elaborate on them largely according to the sequence of the paper.

1.Title: The title of this article is too long. I can not realize the content of this study through the title.

2.The introduction lack of enlightenment on research background and problem statement. The author proposes three examples (Chinese paper enterprise, Volkswagen, and Formosa Plastics Group) of environmental management, but there are weak connections between research focus and the three examples.

3.After reading the Intro, I have no idea about (1) research gaps and how they are identified, (2) overarching theory, (3) why the current study is necessary and important, and (4) where the current study makes a contribution to.

4.The authors need to make a better positioning of their study, This study focuses on the theoretical perspectives of expectancy-valence theory, organizational identity theory, and psychological

I really thank all the affirmative comments from reviewer. That is a very favorable suggestion.

I am revising the review opinion as follows:

1.Title: According to the research content, we have revised the title to "Corporate Sustainable Development Strategy: Effect of Green Shared Vision on Organization Members’ Behavior".

2.Three examples of environmental management firms have been revised. Please refer to the blue note on page 1.

3.I have revised the introduction according to the comments and refer to related literature. Please refer to the blue note on page 3.

4.I have rewritten the sentence according to the comments. Please refer to the blue note on page 3.

【Reviewer 1’s Comment 1】:

【Authors’ Response】:

ownership theory. The author develops a comprehensive model to verify the direct and indirect effects of the GSV on green organizational identity (GOI), green product psychological ownership (GPPO), and OCBE. My suggestion is to rewrite this sentence and explores the relationship between each factor. Authors need to provide a better justification for conducting their study, as well as state clearly the purpose, objectives, and contributions of it.

5.My main concern is that the article presents nothing new. The final section provides only a summary of the findings than a fruitful discussion and insightful conclusions. In addition, there are no theoretical and managerial implications. The overall communication style is not good enough.

6.I'd strongly suggest seeking editorial help as grammar and usage errors throughout the paper are limiting clarity. In the present form, the work is not good enough.

5.I have rewritten the conclusions and implications according to the related literature. Please refer to the revised manuscript on page 12-13.

6.The grammar and usage of the paper have been edited and revised, and proof of editing is attached.

Reviewer 2 Report

The concept of green shared vision has been introduced in green organization management and sustainability research a couple of years ago and since then received a lot of attention. However, we need studies that conceptually refine the concept but also that test the construct in different empirical settings. This study is in line with such development and in that sense the study is interesting and could contribute to our knowledge on green shared vision.

I have some minor concerns regarding the paper and I will elaborate on these concerns below:

Format: The author needs to check and correct the format carefully. For example, the subtitle format of Hypothesis 1 to 8 is inconsistent. Besides, the Figure 2 is not clear, I strongly recommend avoiding showing the numbers and marks on lines and arrows, which covers some important evidences of the outcomes

Title: Please check and correct the English grammar. The title should consider “exploring” instead of “explore”.

Abstract: The dependent variable should play the most important role in the research, and the abstract needs to provide the explanations of how important it is. The abstract only demonstrates the necessary of green shared vision which neglects the vital reason of studying the organizational citizenship behavior for the environment (OCBE). So, I strongly recommend explaining more about OCBE.

Introduction section: This article's introduction section may be too long or excessively detailed. So, I strongly recommend keeping short and to the point.

Literature Review and Hypothesis Development: Again, each subtitle of this section is too long, maybe over 15 characters so the titles should be shortened. Compare to other hypothesis development, the H6 and H8 are too brief. So, the author should consider expanding the development of H6 and H8 or shortening the rest of hypothesis.

Method: The footnote of Table 4 is incorrect which neglects the explanation of ***.

Conclusions and Implications: The most impart part of conclusion is the meaning and contribution of research. So the authors should avoid presenting the limitations at the beginning of this section. Besides, this article should provide more emphasis on the unique contribution and should cite more important papers to dialog with.

Author Response

The response to the Reviewer 2s  comments

Manuscript ID:

ijerph-740402

Title:

Corporate strategies for sustainable consumption: explore the significant motives and antecedents of organizational citizenship behavior for the environment in corporate environmental management

Corporate Sustainable Development Strategy: Effect of Green Shared Vision on Organization Members’ Behavior

【Reviewer 2’s Comment 2】:

【Authors’ Response】:

Comments and Suggestions for Authors

The concept of green shared vision has been introduced in green organization management and sustainability research a couple of years ago and since then received a lot of attention. However, we need studies that conceptually refine the concept but also that test the construct in different empirical settings. This study is in line with such development and in that sense the study is interesting and could contribute to our knowledge on green shared vision.

I have some minor concerns regarding the paper and I will elaborate on these concerns below:

1.Format: The author needs to check and correct the format carefully. For example, the subtitle format of Hypothesis 1 to 8 is inconsistent. Besides, the Figure 2 is not clear, I strongly recommend avoiding showing the numbers and marks on lines and arrows, which covers some important evidences of the outcomes.

2.Title: Please check and correct the English grammar. The title should consider “exploring” instead of “explore”.

3.Abstract: The dependent variable should play the most important role in the research, and the abstract needs to provide the explanations of how important it is. The abstract only demonstrates the necessary of

I really thank all the affirmative comments from reviewer. That is a very favorable suggestion.

I am revising the review opinion as follows:

1.Format: Hypothesis 1 to 8 and Figure 2 have been revised. Please refer to the blue note on page 4-6,10.

2.Title: According to the research content, we have revised the title to "Corporate sustainable development strategy: exploring the impact of green shared vision on members behavior".

3. Abstract: I have revised the abstract. Please refer to the blue note on page 1.

【Reviewer 2’s Comment 2】:

【Authors’ Response】:

green shared vision which neglects the vital reason of studying the organizational citizenship behavior for the environment (OCBE). So, I strongly recommend explaining more about OCBE.

4.Introduction section: This article's introduction section may be too long or excessively detailed. So, I strongly recommend keeping short and to the point.

5.Literature Review and Hypothesis Development: Again, each subtitle of this section is too long, maybe over 15 characters so the titles should be shortened. Compare to other hypothesis development, the H6 and H8 are too brief. So, the author should consider expanding the development of H6 and H8 or shortening the rest of hypothesis.

6. Method: The footnote of Table 4 is incorrect which neglects the explanation of ***.

7. Conclusions and Implications: The most impart part of conclusion is the meaning and contribution of research. So the authors should avoid presenting the limitations at the beginning of this section. Besides, this article should provide more emphasis on the unique contribution and should cite more important papers to dialog with.

4.Introduction section: I have revised the introduction. Please refer to the blue note on page 1-3.

5.Literature Review and Hypothesis Development: I have revised and shortened H1 and H2. Please refer to the blue note on page 3-4.

6.Method: I have revised the footnote of Table 4. Please refer to the blue note on page 9.

7.Conclusions and Implications: I have revised the sentences according to related literature. Please refer to the blue note on page 12-13.

Reviewer 3 Report

The English expression could be improved and does get in the way of the clarity of the arguments.

I am not sure that the examples of firms being bad highlights environmental awareness d the examples were rather egregious actions.

The paper does not focus at all on consumption and thus this needs to be removed from the title. I am also not really sure that is within an environmental management framework, as there is really no variables in the mode that relate to environmental management. These relate to organisational identity, green product psychological ownership, and green shared values. Thus I thin framing the research needs to be stated better.

The authors need to clearly define OCBE as they talk about environmental management, but OECB is supposed to be extra activities that are not requires or managed by the organisation, thus clarity as to how it related to organisational strategy needs to be defined and explained.

The discussion of expectancy-valence theory needs to be expanded, as there the model does not really include rewards to the employees, as the model has the additional effort (OECBE) as the outcomes not that, thus I am unsure where the expectancy- valence link applies?  

It is unclear who in the organisation responded to the survey? I had initially though it would be individual workers, but it read as if it one respondent per organisation this discussion needs to be clearer. It says it is regular employees, but how are these then accessed? There is also NO information on the respondents (i.e. age education role, involvement in environmental activities in the firm, etc. This is an important issue to understand whose views are being assessed.

Figure 2 uses not need the items and as they assessed the items why not use the calculated composite scores of constructs?

There are better ways to assess the mediation effects using methods, using Process?

The Implications focus on activities within the firm, further supporting the fact that sustainable consumption is not a focus of the study. There does need to be more discussion of the theoretical implications of the results.

Author Response

The response to the Reviewer 3s  comments

Manuscript ID:

ijerph-740402

Title:

Corporate strategies for sustainable consumption: explore the significant motives and antecedents of organizational citizenship behavior for the environment in corporate environmental management

Corporate Sustainable Development Strategy: Effect of Green Shared Vision on Organization Members’ Behavior

【Reviewer 3’s Comment 3】:

【Authors’ Response】:

Comments and Suggestions for Authors

The English expression could be improved and does get in the way of the clarity of the arguments.

1.I am not sure that the examples of firms being bad highlights environmental awareness d the examples were rather egregious actions.

2. The paper does not focus at all on consumption and thus this needs to be removed from the title. I am also not really sure that is within an environmental management framework, as there is really no variables in the mode that relate to environmental management. These relate to organisational identity, green product psychological ownership, and green shared values. Thus I thin framing the research needs to be stated better.

3.The authors need to clearly define OCBE as they talk about environmental management, but OECB is supposed to be extra activities that are not requires or managed by the organisation, thus clarity as to how it related to organisational strategy needs to be defined and explained.

4.The discussion of expectancy-valence theory needs to be expanded, as there the model does not really include rewards to the employees, as the model has

Thanks for the reviewer’s comments.

The grammar and usage of the paper have been edited and revised, and proof of editing is attached.

I am revising the review opinion as follows:

1.Three examples of environmental management firms have been revised. Please refer to the blue note on page 1-2.

2. According to the research content, we have revised the title to "Corporate Sustainable Development Strategy: Effect of Green Shared Vision on Organization Members’ Behavior".

3.I have revised the introduction, literature review, hypothesis development, and conclusions according to the research content. Please refer to the blue note on page 1-6,12,13.

4.The internal and external expectation motive of the expectancy-valence theory is added in the research restrictions and future research directions. Please refer to the blue note on

【Reviewer 2’s Comment 2】:

【Authors’ Response】:

the additional effort (OECBE) as the outcomes not that, thus I am unsure where the expectancy- valence link applies?

5. It is unclear who in the organisation responded to the survey? I had initially though it would be individual workers, but it read as if it one respondent per organisation this discussion needs to be clearer. It says it is regular employees, but how are these then accessed? There is also NO information on the respondents (i.e. age education role, involvement in environmental activities in the firm, etc. This is an important issue to understand whose views are being assessed.

6.Figure 2 uses not need the items and as they assessed the items why not use the calculated composite scores of constructs?

7. There are better ways to assess the mediation effects using methods, using Process?

8. The Implications focus on activities within the firm, further supporting the fact that sustainable consumption is not a focus of the study. There does need to be more discussion of the theoretical implications of the results.

page 12.

(1) The research samples focused on the company perspective and did not discuss demographic variables in detail. Future studies can further verify related variables include age, education, etc. I have incorporated your valuable suggestions into the limitations and future of the study. Please refer to the blue note on page 12-13.

(2)The research sample mainly focuses on 8 types of enterprises such as electronics, information services, components manufacturing, computer and peripheral products, electronic products and components, communication equipment manufacturing, machinery and equipment manufacturing, and software industries. The respondents were the company's formal frontline R & D and sales employees. I have revised the section and please refer to the blue note on page 7.

6.Figure 2 is full model results, model correlation index and load of each factor. I have noted in Figure 2. Please refer to the blue note on page 9-10.

7. Baron & Kenny (1986) verification method or Preacher & Hayes (2008) bootstrapping method is commonly used to verify the effect of the mediator. Please refer to the blue note on page 10-11.

8. I have revised the Implications according to related literature. Please refer to the blue note on page 12-13.

Round 2

Reviewer 1 Report

The authors improved a lot the quality level of the manuscript. This paper addresses a very interesting topic. Therefore, the paper has enough novel content to be ready for publication.